# GraphLearner: Graph Node Clustering with Fully Learnable Augmentation

## ABSTRACT

Contrastive deep graph clustering (CDGC) leverages the power of contrastive learning to group nodes into different clusters. The quality of contrastive samples is crucial for achieving better performance, making augmentation techniques a key factor in the process. However, the augmentation samples in existing methods are always predefined by human experiences, and agnostic from the downstream task clustering, thus leading to high human resource costs and poor performance. To overcome these limitations, we propose a Graph Node Clustering with Fully Learnable Augmentation, termed **GraphLearner**. It introduces learnable augmentors to generate high-quality and task-specific augmented samples for CDGC. GraphLearner incorporates two learnable augmentors specifically designed for capturing attribute and structural information. Moreover, we introduce two refinement matrices, including the high-confidence pseudo-label matrix and the cross-view sample similarity matrix, to enhance the reliability of the learned affinity matrix. During the training procedure, we notice the distinct optimization goals for training learnable augmentors and contrastive learning networks. In other words, we should both guarantee the consistency of the embeddings as well as the diversity of the augmented samples. To address this challenge, we propose an adversarial learning mechanism within our method. Besides, we leverage a two-stage training strategy to refine the high-confidence matrices. Extensive experimental results on six benchmark datasets validate the effectiveness of GraphLearner.

## CCS CONCEPTS

• **Theory of computation** → **Unsupervised learning and clustering**; • **Computing methodologies** → **Cluster analysis**.

## KEYWORDS

Graph Node Clustering, Graph Neural Networks, Learnable Augmentation

## 1 INTRODUCTION

In recent years, graph learning methods have attracted considerable attention in various multimedia applications, e.g., node classification [18], clustering [14, 32, 37], etc. Among all directions, deep graph clustering, which aims to encode nodes with neural networks

*ACM MM, 2024, Melbourne, Australia*

© 2024 Copyright held by the owner/author(s). Publication rights licensed to ACM.
ACM ISBN 978-x-xxxx-xxxx-x/YY/MM
https://doi.org/10.1145/nnnnnnn.nnnnnnn

and divide them into disjoint clusters without manual labels, has become a hot auxiliary task in information systems.

With the strong capability of capturing implicit supervision, contrastive learning has become an important technique in deep graph clustering. In general, the existing methods first generate augmented graph views by perturbing node connections or attributes, and then keep the same samples in different views consistent while enlarging the difference between distinct samples. Although verified effective, we find that the performance of the existing graph contrastive clustering methods [15, 43] heavily depends on the augmented view. However, the existing augmentation methods are usually predefined and selected with a cumbersome search. The connection of augmentation and the specific downstream task is deficient. To alleviate this problem, in graph classification, JOAO [39] selects a proper augmentation type among several predefined candidates. Although better performance is achieved, the specific augmentation process is still based on the predefined schemes and cannot be optimized by the network. To fill this gap, AD-GCL [23] proposes a learnable augmentation scheme to drop edges according to Bernoulli distribution, while neglecting augmentations on node attributes. More recently, AutoGCL [38] proposes an auto augmentation strategy to mask or drop nodes via learning a probability distribution. A large step is made by these algorithms by proposing learnable augmentation. However, these strategies only focus on exploring augmentation over affinity matrices while neglecting the learning of good attribute augmentations. Moreover, previous methods isolate the representation learning process with the specific downstream tasks, making the learned representation less suitable for the final learning task, degrading the algorithm performance.

To solve this issue, we propose a fully learnable augmentation strategy for deep contrastive clustering, which generates more suitable augmented views. Specifically, we design the learnable augmenters to learn the structure and attribute information dynamically, thus avoiding the carefully selections of the existing and predefined augmentations. Besides, to improve the reliability of the learned structure, we refine that with the high-confidence clustering pseudo-label matrix and the cross-view sample similarity matrix. Moreover, an adversarial learning mechanism is proposed to learn the consistency of embeddings in latent space, while keeping the diversity of the augmented view. Lastly, during the model training, we present a two-stage training strategy to obtain high-confidence refinement matrices. We summarize the properties of the existing graph augmentation algorithms in Table. 1. From the results, we could observe that our proposed method offers a more comprehensive approach.

By those settings, the augmentation strategies do not rely on tedious manual trial-and-error and repetitive attempts. Moreover, we enhance the connection between the augmentation and the clustering task and integrate the clustering task and the augmentation learning into the unified framework. Firstly, the high-quality

**Table 1: An overview of graph augmentation methods. "S" and "A" denotes the structure augmentation and attribution augmentation, respectively. Besides, "Opt" means that the augmentation is optimized by the downstream task.**

| Method | S | A | Opt | Type | Task |
|--------|---|---|-----|------|------|
| JOAO | ✓ | ✓ | ✗ | Predefined | Classification |
| AutoGCL | ✓ | ✓ | ✗ | Predefined | Classification |
| AD-GCL | ✓ | ✗ | ✗ | Predefined | Classification |
| CCGC | ✗ | ✓ | ✗ | Learnable | Clustering |
| CONVERT | ✗ | ✓ | ✗ | Learnable | Clustering |
| Ours | ✓ | ✓ | ✓ | Learnable | Clustering |

augmented graph improves the discriminative capability of embeddings, thus better assisting the clustering task. Meanwhile, the high-confidence clustering results are utilized to refine the augmented graph structure. Concretely, samples within the same clusters are more likely to link, while edges between samples from different clusters are removed.

The key contributions of this paper are listed as follows:

- By designing the structure and attribute augmentor, we propose a fully learnable data augmentation framework for deep contrastive graph clustering termed GraphLearner to dynamically learn the structure and attribute information.
- We refine the augmented graph structure with the cross-view similarity matrix and high-confidence pseudo-label matrix to improve the reliability of the learned affinity matrix. Thus, the clustering task and the augmentation learning are integrated into the unified framework and promote each other.
- Extensive experimental results have demonstrated that our method outperforms the existing state-of-the-art deep graph clustering competitors.

## 2 METHOD

In this section, we propose a novel attribute graph contrastive clustering method with fully learnable augmentation (GraphLearner). The overall framework of GraphLearner is shown in Fig.1. The main components of the proposed method include the fully learnable augmentation module and the dual refinement module. We will detail the proposed GraphLearner in the following subsections.

### 2.1 Notations Definition

For an undirected graph $\mathbf{G} = \{\mathbf{X}, \mathbf{A}\}$. $\mathbf{X} \in \mathbb{R}^{N \times D}$ is the attribute matrix, and $\mathbf{A} \in \mathbb{R}^{N \times N}$ represents the original adjacency matrix. $\mathbf{D} = diag(d_1, d_2, \ldots, d_N) \in \mathbb{R}^{N \times N}$ is denoted as the degree matrix, where $d_i = \sum_{(v_i, v_j) \in \mathcal{E}} a_{ij}$. We define the graph Laplacian matrix as $\mathbf{L} = \mathbf{D} - \mathbf{A}$. With the help of renormalization trick $\widetilde{\mathbf{A}} = \mathbf{A} + \mathbf{I}$, the normalized graph Laplacian matrix is denoted as $\widetilde{\mathbf{L}} = \widehat{\mathbf{D}}^{-\frac{1}{2}} \widehat{\mathbf{L}} \widehat{\mathbf{D}}^{-\frac{1}{2}}$. Moreover, we define $sim(\cdot)$ as a non-parametric metric function to calculate pair-wise similarity, e.g., cosine similarity function. $\mathbf{Aug}_S$ and $\mathbf{Aug}_X$ represent the augmented structure and attribute matrix, respectively. The basic notations are summarized in Table 2.

**Table 2: Notation summary.**

| Notation | Meaning |
|----------|---------|
| $\mathbf{X} \in \mathbb{R}^{N \times D}$ | Attribute matrix |
| $\mathbf{A} \in \mathbb{R}^{N \times N}$ | Original adjacency matrix |
| $\mathbf{D} \in \mathbb{R}^{N \times N}$ | Degree matrix |
| $\mathbf{F}^{v_k} \in \mathbb{R}^{N \times d}$ | Node embeddings in $k$-th view |
| $sim(\cdot)$ | Non-parametric metric function |
| $\mathbf{S} \in \mathbb{R}^{N \times N}$ | Similarity sample matrix |
| $\mathbf{Z} \in \mathbb{R}^{N \times N}$ | High-confidence pseudo label matrix |
| $\mathbf{Aug}_X \in \mathbb{R}^{N \times d}$ | Augmented attribute matrix |
| $\mathbf{Aug}_S \in \mathbb{R}^{N \times N}$ | Augmented structure matrix |

### 2.2 Fully Learnable Augmentation Module

Different from previously augmentation method, in this subsection, we propose a fully learnable graph augmentation strategy in both structure and attribute level. To be specific, we design the structure augmentor and attribute augmentor to dynamically learn the structure and attribute, respectively. In the following, we will introduce these augmentors in detail.

*2.2.1 Structure Augmentor.* We design the structure augmentor to obtain the structure of the augmented view. Specifically, we propose three types structure augmentor, i.e., MLP-based structure augmentor, GCN-based structure augmentor, and Attention-based augmentor.

**MLP-based Structure Augmentor.** We use Multi-Layer Perception (MLPs) to generate the structure $\mathbf{Aug}_S$. This procedure can be represented as follows:

$$\mathbf{F} = MLP(A), \ \mathbf{A}_{MLP} = sim(\mathbf{F} \cdot \mathbf{F}^{\mathrm{T}}), \tag{1}$$

where $\mathbf{F} \in \mathbb{R}^{N \times D}$ is the embedding of the original adjacency. We adopt the cosine similarity function as $sim(\cdot)$ to calculate the similarity of $\mathbf{F}$. The calculated similarity matrix can be regarded as the learned structure matrix $\mathbf{A}_{MLP}$.

**GCN-based Structure Augmentor** is the second type of our designed structure generator, which embeds the attribute matrix $\mathbf{X}$ and original adjacency matrix $\mathbf{A}$ into embeddings in the latent space. For simplicity, we define the GCN-based structure augmentor as:

$$\mathbf{F} = \sigma(\widetilde{\mathbf{D}}^{-\frac{1}{2}} \widetilde{\mathbf{A}} \widetilde{\mathbf{D}}^{-\frac{1}{2}} \mathbf{X}), \ \mathbf{A}_{GCN} = sim(\mathbf{F} \cdot \mathbf{F}^{\top}), \tag{2}$$

where $\mathbf{F}$ is embedding extracted by the GCN-based network, for example, GCN [12], GCN-Cheby [6], etc. $\sigma(\cdot)$ is a non-linear operation.

**Attention-based Structure Augmentor.** Inspire by GAT [27], we design an attentive network to capture the important structure of the input graph $\mathbf{G}$. To be specific, the normalized attention coefficient matrix $\mathbf{A}_{att_{ij}}$ between node $x_i$ and $x_j$ could be computed as:

$$\mathbf{A}_{att_{ij}} = \vec{\mathbf{n}}^{\top}(\mathbf{W}_{x_i} || \mathbf{W}_{x_j}),$$
$$\mathbf{A}_{att_{ij}} = \frac{e^{(\mathbf{A}_{att_{ij}})}}{\sum_{k \in \mathcal{N}_i} e^{(\mathbf{A}_{att_{ik}})}}, \tag{3}$$

where $\vec{\mathbf{n}}$ and $\mathbf{W}$ is the learnable weight vector and weight matrix, respectively. || is the concatenation operation between the weight

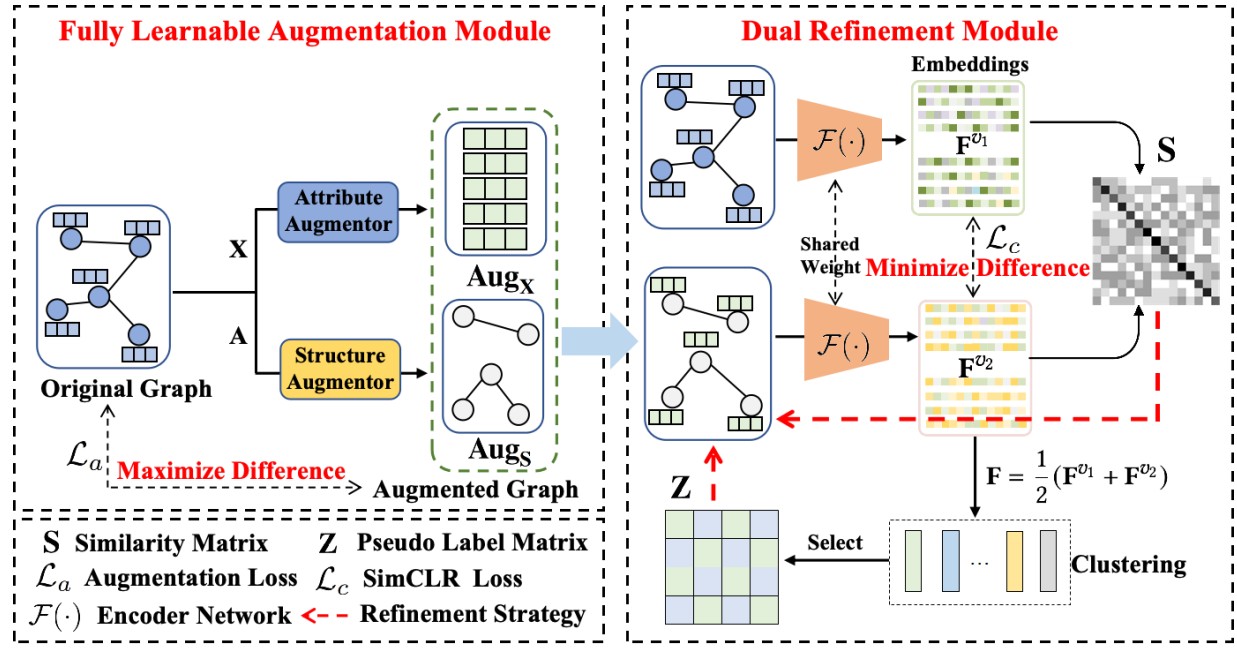

**Figure 1: Illustration of the fully learnable augmentation algorithm for attribute graph contrastive clustering. In our proposed algorithm, we design the learnable augmentors to to dynamically learn the structure and attribute information. Besides, we optimize the structure of the augmented view with two aspects, i.e., high-confidence clustering pseudo label matrix and cross-view similarity matrix, which integrates the clustering task and the augmentation learning into the unified framework. Moreover, we propose an adversarial learning mechanism to keep cross-view consistency in the latent space while ensuring the diversity of augmented views. Lastly, a two-stage training strategy is designed to obtain high-confidence refinement matrices, thus improving the reliability of the learned graph structure.**

matrix $\mathbf{W}_{x_i}$ and $\mathbf{W}_{x_j}$, and $\mathcal{N}_i$ represents the indices the neighbors of node $x_i$. By this setting, the model could preserve important topological and semantic graph patterns via the attention mechanism.

*2.2.2 Attribute Augmentor.* To make the augmented view in a fully learnable manner, we design the attribute augmentor to dynamically learn the original attribute. Specifically, we design two types attribute augmentor, i.e., MLP-based attribute augmentor and attention-based attribute augmentor.

**MLP-based Attribute Augmentor.** Similar to the MLP-based structure augmentor, we utilize the Multi-Layer Perception (MLP) as the network to learn the original attribute matrix $\mathbf{X}$. The learned attribute matrix $\mathbf{Aug_X} \in \mathbb{R}^{N \times D}$ can be presented as:

$$\mathbf{X}_{MLP} = MLP(\mathbf{X}). \qquad (4)$$

where $MLP(\cdot)$ is the MLP network to learn the attribute.

**Attention-based Attribute Augmentor.** To guide the network to take more attention to the important node attributes, we design an attention-based attribute augmentor. Specifically, we map the node attributes into three different latent spaces:

$$\begin{aligned} \mathbf{Q} &= \mathbf{W}_q \mathbf{X}^\top \\ \mathbf{K} &= \mathbf{W}_k \mathbf{X}^\top \\ \mathbf{V} &= \mathbf{W}_v \mathbf{X}^\top \end{aligned} \qquad (5)$$

where $\mathbf{W}_q \in \mathbb{R}^{D \times D}, \mathbf{W}_k \in \mathbb{R}^{D \times D}, \mathbf{W}_v \in \mathbb{R}^{D \times D}$ are the learnable parameter matrices. And $\mathbf{Q} \in \mathbb{R}^{D \times N}, \mathbf{K} \in \mathbb{R}^{D \times N}$ and $\mathbf{V} \in \mathbb{R}^{D \times N}$ denotes the query matrix, key matrix and value matrix, respectively.

The attention-based attribute matrix $\mathbf{Aug_X}$ can be calculated by:

$$\mathbf{Aug_X} = softmax(\frac{\mathbf{K}^\top \mathbf{Q}}{\sqrt{D}})\mathbf{V}^\top, \qquad (6)$$

After the structure augmentor and attribute augmentor, we could obtain the augmented view $\mathbf{G}' = (\mathbf{Aug_S}, \mathbf{Aug_X})$. The , which is fully learnable.

## 2.3 Dual Refinement Module

In this subsection, we propose a dual refinement module to optimize the learned graph structure. To be specific, the cross-view sample similarity matrix and the high-confidence matrix are generated to improve the quality of the structure in augmented view. Firstly, we use encoder network $\mathcal{F}(\cdot)$ to obtain the embeddings of the original view $\mathbf{G}$ and the augmented view $\mathbf{G}'$ with $\ell^2$-norm as follows:

$$\begin{aligned} \mathbf{F}^{v_1} &= \mathcal{F}(\mathbf{G}), \\ \mathbf{F}^{v_2} &= \mathcal{F}(\mathbf{G}'). \end{aligned} \qquad (7)$$

In the following, we fuse the two views of the node embeddings as follows:

$$\mathbf{F} = \frac{1}{2}(\mathbf{F}^{v_1} + \mathbf{F}^{v_2}). \tag{8}$$

Then we perform K-means [9] on $\mathbf{F}$ and obtain the clustering results. After that, we will refine the learned view in two manners, i.e., similarity matrix and pseudo labels matrix refinement.

**Similarity Matrix Refinement.** Through $\mathcal{F}(\cdot)$, we could obtain the embeddings of each view. Subsequently, the similarity matrix $\mathbf{S}$ represents the similarity between $i$-th sample in the first view and $j$-th sample in the second view as formulated:

$$\mathbf{S} = \frac{\left\langle \mathbf{F}^{v_1} \cdot (\mathbf{F}^{v_2})^\top \right\rangle}{||\mathbf{F}^{v_1}||_2 \cdot ||\mathbf{F}^{v_2}||_2}, \tag{9}$$

where $\mathbf{S}$ is the cross-view similarity matrix, and $\langle \cdot \rangle$ is the function to calculate similarity. Here, we adopt cosine similarity. The proposed similarity matrix $\mathbf{S}$ measures the similarity between samples by comprehensively considering attribute and structure information. The connected relationships between different nodes could be reflected by $\mathbf{S}$. Therefore, we utilize $\mathbf{S}$ to refine the structure in augmented view with Hadamard product:

$$\mathbf{Aug_S} = \mathbf{Aug_S} \odot \mathbf{S}. \tag{10}$$

**Pseudo Labels Matrix Refinement.** To further improve the reliability of the learned structure matrix, we extract reliable clustering information to construct the matrix to further refine the structure in augmented view. Concretely, we utilize the top $\tau$ high-confidence pseudo labels $\mathbf{p}$ to construct the matrix as follows:

$$\mathbf{Z}_{ij} = \begin{cases} 1 & \mathbf{p}_i = \mathbf{p}_j, \\ 0 & \mathbf{p}_i \neq \mathbf{p}_j, \end{cases} \tag{11}$$

where $\mathbf{Z}_{ij}$ denotes the category relation between $i$-th and $j$-th samples. In detail, when $\mathbf{Z}_{ij} = 1$, two samples have the same pseudo label. While $\mathbf{Z}_{ij} = 0$ implies that two samples have different pseudo labels. The pseudo-label matrix is constructed by the high-confidence category information. Therefore, the adjacency relation in the graph could be well reflected, leading to optimizing the structure of the learned structure in the augmented view. The pseudo labels matrix refines the learned structure with Hadamard product as:

$$\mathbf{Aug_S} = \mathbf{Aug_S} \odot \mathbf{Z}. \tag{12}$$

In summary, in this subsection, we propose two strategies to refine the structure of the augmented view. Firstly, $\mathbf{S}$ is calculated by the cross-view similarity. The value of $\mathbf{S}$ represents the probability of connection relationships of the nodes. The structure of $\mathbf{Aug_S}$ is optimized by $\mathbf{S}$ in the training process. Besides, we utilize the high-confidence clustering pseudo labels to construct the reliable node connection, which is constructed when the node belongs to the same category. By those settings, the learned structure is regularized by the similarity matrix and the high-confidence matrix, thus improving the reliability of the structure in the augmented view. Moreover, the connection between augmentation and the clustering task is enhanced.

---

**Algorithm 1 GraphLearner**

---

**Input**: The input graph $\mathbf{G} = \{\mathbf{X}, \mathbf{A}\}$; The iteration number $I$; Hyper-parameters $\tau, \alpha$.
**Output**: The clustering result $\mathbf{R}$.

1: **for** $i = 1$ to $I$ **do**
2:      Obtain the learned structure matrix $\mathbf{Aug_S}$ and attribute matrix $\mathbf{Aug_X}$ with our augmentors.
3:      Encode the node with the network $\mathcal{F}(\cdot)$ to obtain the node embeddings $\mathbf{F}^{v_1}$ and $\mathbf{F}^{v_2}$.
4:      Fuse $\mathbf{F}^{v_1}$ and $\mathbf{F}^{v_2}$ to obtain $\mathbf{F}$ with Eq. (8).
5:      Perform K-means on $\mathbf{F}$ to obtain the clustering result.
6:      Calculate the similarity matrix of $\mathbf{F}^{v_1}$ and $\mathbf{F}^{v_2}$.
7:      Obtain high-confidence pseudo label matrix.
8:      Refine the learned structure matrix $\mathbf{Aug_S}$ with Eq.(10) and Eq. (12).
9:      Calculate the learnable augmentation loss $\mathcal{L}_a$ with Eq. (13).
10:     Calculate the contrastive loss $\mathcal{L}_c$ with Eq. (14).
11:     Update the whole network by minimizing $\mathcal{L}$ in Eq. (15).
12: **end for**
13: Perform K-means on $\mathbf{F}$ to obtain the final clustering result $\mathbf{R}$.

---

## 2.4 Loss Function

The proposed GraphLearner framework follows the common contrastive learning paradigm, where the model maximizes the agreement of the cross-view [45]. In detail, GraphLearner jointly optimizes two loss functions, including the learnable augmentation loss $\mathcal{L}_a$ and the contrastive loss $\mathcal{L}_c$.

To be specific, $\mathcal{L}_a$ is the Mean Squared Error (MSE) loss between the original graph $\mathbf{G} = \{\mathbf{X}, \mathbf{A}\}$ and the learnable graph $\mathbf{G}' = \{\mathbf{Aug_X}, \mathbf{Aug_S}\}$, which can be formulated as:

$$\mathcal{L}_a = -(\left\| \mathbf{A} - \mathbf{Aug_S} \right\|_2^2 + \left\| \mathbf{X} - \mathbf{Aug_X} \right\|_2^2). \tag{13}$$

In GraphLearner, we utilize the normalized temperature-scaled cross-entropy loss (NT-Xent) to pull close the positive samples, while pushing the negative samples away. The contrastive loss $\mathcal{L}_c$ is defined as:

$$\begin{aligned} l_i &= -log \frac{\exp(\text{sim}(\mathbf{F}_i^{v_1}, \mathbf{F}_i^{v_2})/\text{temp})}{\sum_{k=1, k \neq i}^{N} \exp(\text{sim}(\mathbf{F}_i^{v_1}, \mathbf{F}_k^{v_2})/\text{temp})}, \\ \mathcal{L}_c &= \frac{1}{N} \sum_{i=1}^{N} l(i), \end{aligned} \tag{14}$$

where temp is a temperature parameter. $\text{sim}(\cdot)$ denotes the function to calculate the similarity, e.g., inner product.

The total loss of GraphLearner is calculated as follows:

$$\mathcal{L} = \mathcal{L}_a + \alpha \mathcal{L}_c, \tag{15}$$

where $\alpha$ is the trade-off between $\mathcal{L}_a$ and $\mathcal{L}_c$. The first term in Eq.(15) encourages the network to generate the augmented view with distinct semantics to ensure the diversity in input space, while the second term is the contrastive paradigm to learn the consistency of two views in latent space. The discriminative capacity of the network could be improved by minimizing the total loss function. The network is optimized by Eq.(15) during the whole training process. The detailed learning process of GraphLearner is shown in Algorithm 1.

**Table 3: Clustering performance on six datasets (mean ± std). Best results are bold values and the second best values are unerlined. OOM denotes out-of-memory during the training process.**

| Dataset | Metric | DAEGC | SDCN | DCRN | AGC-DRR | CCGC | CONVERT | SUBLIME | GCA | AFGRL | AutoSSL | Ours |
|---|---|---|---|---|---|---|---|---|---|---|---|---|
| CORA | ACC | 70.43±0.36 | 35.60±2.83 | 61.93±0.47 | 40.62±0.55 | 73.01±1.11 | 73.21±1.23 | 71.14±0.74 | 53.62±0.73 | 26.25±1.24 | 63.81±0.57 | **74.91±1.78** |
| | NMI | 52.89±0.69 | 14.28±1.91 | 45.13±1.57 | 18.74±0.73 | 55.78±0.57 | 54.34±0.13 | 53.88±1.02 | 46.87±0.65 | 12.36±1.54 | 47.62±0.45 | **58.16±0.83** |
| | ARI | 49.63±0.43 | 07.78±3.24 | 33.15±0.14 | 14.80±1.64 | 51.45±0.75 | 50.01±2.12 | 50.15±0.14 | 30.32±0.98 | 14.32±1.87 | 38.92±0.77 | **53.82±2.25** |
| | F1 | 68.27±0.57 | 24.37±1.04 | 49.50±0.42 | 31.23±0.57 | 70.45±1.73 | 71.33±1.09 | 63.11±0.58 | 45.73±0.47 | 30.20±1.15 | 56.42±0.21 | **73.33±1.86** |
| AMAP | ACC | 75.96±0.23 | 53.44±0.81 | OOM | 76.81±1.45 | 76.44±0.48 | 76.34±0.65 | 27.22±1.56 | 56.81±1.44 | 75.51±0.77 | 54.55±0.97 | **77.24±0.87** |
| | NMI | 65.25±0.45 | 44.85±0.83 | OOM | 66.54±1.24 | 66.78±0.71 | 65.48±0.87 | 06.37±1.89 | 48.38±2.38 | 64.05±0.15 | 48.56±0.71 | **67.12±0.92** |
| | ARI | 57.12±0.24 | 31.21±1.23 | OOM | **60.15±1.56** | 56.45±0.87 | 57.48±1.24 | 05.36±2.14 | 26.85±0.44 | 54.45±0.48 | 26.87±0.34 | 58.14±0.82 |
| | F1 | 69.87±0.54 | 50.66±1.49 | OOM | 71.03±0.64 | 71.57±0.94 | 72.48±0.61 | 15.97±1.53 | 53.59±0.57 | 69.99±0.34 | 54.47±0.83 | **73.02±2.34** |
| BAT | ACC | 52.67±0.00 | 53.05±4.63 | 67.94±1.45 | 47.79±0.02 | 75.00±0.44 | 74.12±1.57 | 45.04±0.19 | 54.89±0.34 | 50.92±0.44 | 42.43±0.47 | **75.50±0.87** |
| | NMI | 21.43±0.35 | 25.74±5.71 | 47.23±0.74 | 19.91±0.24 | 49.15±0.67 | 50.01±0.25 | 22.03±0.48 | 38.88±0.23 | 27.55±0.62 | 17.84±0.98 | **50.58±0.90** |
| | ARI | 18.18±0.29 | 21.04±4.97 | 39.76±0.87 | 14.59±0.13 | 46.45±0.17 | **50.66±1.97** | 14.45±0.87 | 26.69±2.85 | 21.89±0.74 | 13.11±0.81 | 47.45±1.53 |
| | F1 | 52.23±0.03 | 46.45±5.90 | 67.40±0.35 | 42.33±0.51 | 73.15±0.43 | 75.02±0.53 | 44.00±0.62 | 53.71±0.34 | 46.53±0.57 | 34.84±0.15 | **75.40±0.88** |
| EAT | ACC | 36.89±0.15 | 39.07±1.51 | 50.88±0.55 | 37.37±0.11 | 56.14±0.22 | 38.80±0.35 | 48.51±1.55 | 37.42±1.24 | | 31.33±0.52 | **57.22±0.73** |
| | NMI | 05.57±0.06 | 08.83±2.54 | 22.01±1.23 | 07.00±0.85 | 31.54±1.55 | 32.14±0.10 | 14.96±0.75 | 28.36±1.23 | 11.44±1.41 | 07.63±0.85 | **33.47±0.34** |
| | ARI | 05.03±0.08 | 06.31±1.95 | 18.13±0.85 | 04.88±0.91 | 24.87±2.23 | 25.14±1.87 | 10.29±0.88 | 19.61±1.25 | 06.57±1.73 | 02.13±0.67 | **26.21±0.81** |
| | F1 | 34.72±0.16 | 33.42±3.10 | 47.06±0.66 | 35.20±0.17 | 56.44±0.74 | 56.47±0.27 | 32.31±0.97 | 48.22±0.33 | 30.53±1.47 | 21.82±0.98 | **57.53±0.67** |
| CITESEER | ACC | 64.54±1.39 | 65.96±0.31 | 69.86±0.18 | 68.32±1.83 | 68.48±0.44 | 67.54±0.58 | 68.25±1.21 | 60.45±1.03 | 31.45±0.54 | 66.76±0.67 | **70.12±0.36** |
| | NMI | 36.41±0.86 | 38.71±0.32 | 42.86±0.35 | 43.28±1.41 | 42.84±1.78 | 40.28±0.63 | 43.15±0.14 | 36.15±0.78 | 15.17±0.47 | 40.67±0.84 | **43.56±0.35** |
| | ARI | 37.78±1.24 | 40.17±0.43 | 43.64±0.30 | 43.34±2.33 | 43.48±0.11 | 40.24±0.46 | 44.21±0.54 | 35.20±0.96 | 14.32±0.78 | 38.73±0.55 | **44.85±0.69** |
| | F1 | 62.20±1.32 | 63.62±0.24 | 64.83±0.21 | 64.82±1.60 | 62.78±0.53 | 63.85±0.85 | 63.12±0.42 | 56.42±0.94 | 30.20±0.71 | 58.22±0.68 | **65.01±0.39** |
| UAT | ACC | 52.29±0.49 | 52.25±1.91 | 49.92±1.25 | 42.64±0.31 | 52.48±2.87 | 53.47±0.16 | 48.74±0.54 | 39.39±1.46 | 41.50±0..25 | 42.52±0.64 | **54.76±1.42** |
| | NMI | 21.33±0.44 | 21.61±1.26 | 24.09±0.53 | 11.15±0.24 | 24.48±0.88 | 24.16±2.13 | 21.85±0.62 | 24. 05±0.25 | 17.33±0.54 | 17.86±0.22 | **25.23±0.96** |
| | ARI | 20.50±0.51 | 21.63±1.49 | 17.17±0.69 | 09.50±0.25 | 18.51±1.57 | 19.40±0.97 | 19.01±0.45 | 14. 37±0.19 | 13.62±0.57 | 13.13±0.71 | **19.44±1.69** |
| | F1 | 50.33±0.64 | 45.59±3.54 | 44.81±0.87 | 35.18±0.32 | 51.87±2.23 | 52.52±0.08 | 46.19±0.87 | 35.72±0.28 | 36.52±0.89 | 34.94±0.87 | **53.61±2.61** |

**Table 4: Dataset information.**

| Dataset | Type | Sample | Dimension | Edge | Class |
|---|---|---|---|---|---|
| CORA | Graph | 2708 | 1433 | 5429 | 7 |
| AMAP | Graph | 7650 | 745 | 119081 | 8 |
| CITESEER | Graph | 3327 | 3703 | 4732 | 6 |
| UAT | Graph | 1190 | 239 | 13599 | 4 |
| BAT | Graph | 131 | 81 | 1038 | 4 |
| EAT | Graph | 399 | 203 | 5994 | 4 |

The memory cost of $\mathcal{L}$ is acceptable. We utilize $B$ to denote the batch size. The dimension of the embeddings is $D$. The time complexity of $\mathcal{L}$ is $O(B^2D)$. And the space complexity of $\mathcal{L}$ is $O(B^2)$ due to matrix multiplication. The detailed experiments are shown in section 3.3. Besides, we design a two-stage training strategy to improve the confidence of the clustering pseudo labels during the overall training procedure. To be specific, the discriminative capacity of the network is improved by the first training stage. Then, in the second stage, we refine the learned structure $\mathbf{Aug_S}$ in the augmented view with the more reliable similarity matrix and the pseudo labels matrix.

## 3 EXPERIMENT

In this section, we implement experiments to verify GraphLearner. The downstream task of our method is graph node clustering, which is implemented in unsupervised scenario. Therefore, the compared methods do not include the graph classification method, e.g., JOAO [39], AutoGCL [38], AD-GCL [23]. We have already described the differences in the Introduction and Related work. The effectiveness and superiority of our proposed GraphLearner can be illustrated by answering the following questions: **RQ1**: How effective is GraphLearner for attribute node clustering? **RQ2**: How about the efficiency about GraphLearner? **RQ3**: How does the proposed module influence the performance of GraphLearner? **RQ4**: How do the hyper-parameters impact the performance of GraphLearner? **RQ5**: What is the clustering structure revealed by GraphLearner?

### 3.1 Experimental Setup

**Benchmark Datasets** The experiments are implemented on six widely-used benchmark datasets, including CORA [5], BAT [19], EAT [19], AMAP [15], CITESEER [1], and UAT [19]. The summarized information is shown in Table 4. Detailed description of the datasets are shown in Section 1 of Appendix.

**Training Details** The experiments are conducted on the Py-Torch deep learning platform with the Intel Core i7-7820x CPU, one NVIDIA GeForce RTX 3080Ti GPU, 64GB RAM. The max training epoch number is set to 400. For fairness, we conduct ten runs for all methods. For the baselines, we adopt their source with original settings and reproduce the results.

**Evaluation Metrics** The clustering performance is evaluated by four metrics, including Accuracy (ACC), Normalized Mutual Information (NMI), Average Rand Index (ARI), and macro F1-score (F1).

**Parameter Setting** In our model, the learning rate is set to 1e-3 for UAT, 1e-4 for CORA/CITESEER, 1e-5 for AMAP/BAT, and 1e-7 for EAT, respectively. The threshold $\tau$ is set to 0.95 for all datasets. The epoch to begin the second training stage is set to 200. The trade-off $\alpha$ is set to 0.5. Due to the limited space, The hyper-parameter settings are summarized in Table. 1 of the Appendix.

### 3.2 Performance Comparison (RQ1)

In this subsection, to verify the superiority of GraphLearner, we compare the clustering performance of our proposed algorithm with 10 baselines on six datasets with four metrics. We divide these methods into four categories, i.e., classical deep clustering methods

[1] http://citeseerx.ist.psu.edu/index

**Table 5: Ablation studies over the learnable graph augmentation module of GraphLearner on six datasets. "(w/o) Aug$_X$", "(w/o) Aug$_S$" and "(w/o) Aug$_X$ & Aug$_S$" represent the reduced models by removing the structure augmentor, the attribute augmentor, and both, respectively. Additionally, our algorithm is compared with four classic data augmentations.**

| Dataset | Metric | (w/o) Aug_X | (w/o) Aug_S | (w/o) Aug_X & Aug_S | Mask Feature | Drop Edges | Add Edges | Diffusion | Ours |
|---|---|---|---|---|---|---|---|---|---|
| CORA | ACC | 65.60±4.95 | 71.92±1.32 | 62.16±2.57 | 70.60±0.91 | 60.29±2.42 | 68.02±1.93 | 72.68±1.00 | **74.91±1.78** |
| | NMI | 48.81±3.95 | 53.82±1.99 | 40.84±1.45 | 53.99±1.48 | 48.40±1.91 | 50.78±1.93 | 55.80±1.22 | **58.16±0.83** |
| | ARI | 42.42±5.09 | 49.20±1.56 | 34.84±2.82 | 47.80±1.09 | 39.78±2.21 | 43.56±1.83 | 50.45±1.24 | **53.82±2.25** |
| | F1 | 60.86±8.35 | 69.82±0.88 | 60.46±3.10 | 69.39±0.85 | 55.40±4.64 | 66.93±1.96 | 69.11±0.78 | **73.33±1.86** |
| AMAP | ACC | 73.18±4.67 | 73.14±1.10 | 68.32±0.98 | 72.73±0.41 | 64.22±4.15 | 74.51±0.16 | 72.99±0.53 | **77.24±0.87** |
| | NMI | 61.27±5.13 | 60.99±0.75 | 53.76±1.12 | 61.99±0.59 | 53.07±3.57 | 62.94±0.28 | 61.57±0.92 | **67.12±0.92** |
| | ARI | 51.89±6.11 | 52.27±1.72 | 44.50±1.28 | 49.98±0.83 | 46.07±3.58 | 53.45±0.32 | 50.64±0.67 | **58.14±0.82** |
| | F1 | 69.19±4.57 | 68.73±1.39 | 63.58±0.93 | 68.36±0.85 | 56.14±5.21 | 69.16±0.14 | 68.16±0.57 | **73.02±2.34** |
| BAT | ACC | 69.01±2.58 | 70.84±2.64 | 64.43±2.35 | 58.85±3.14 | 53.28±2.60 | 66.03±3.19 | 56.95±3.63 | **75.50±0.87** |
| | NMI | 46.52±1.00 | 47.96±2.04 | 40.98±2.30 | 38.04±2.80 | 28.44±2.01 | 41.05±3.20 | 37.79±4.76 | **50.58±0.90** |
| | ARI | 42.16±1.43 | 42.97±2.08 | 35.19±2.96 | 25.67±4.52 | 20.86±2.67 | 36.03±4.28 | 29.43±3.67 | **47.45±1.53** |
| | F1 | 67.05±4.28 | 70.03±3.71 | 63.08±3.08 | 57.94±3.94 | 52.27±3.00 | 65.09±3.15 | 49.84±4.73 | **75.40±0.88** |
| EAT | ACC | 56.42±1.57 | 55.31±0.88 | 38.80±1.67 | 50.13±2.11 | 47.19±1.81 | 40.03±5.50 | 45.56±1.86 | **57.22±0.73** |
| | NMI | 33.11±1.34 | 32.65±1.02 | 12.06±2.35 | 25.74±2.54 | 28.25±4.64 | 09.01±7.18 | 21.12±3.12 | **33.47±0.34** |
| | ARI | **26.66±0.95** | 25.70±0.86 | 07.72±2.55 | 18.46±2.48 | 22.37±4.43 | 07.72±6.23 | 16.28±4.00 | 26.21±0.81 |
| | F1 | 56.36±1.99 | 54.51±2.79 | 31.53±2.62 | 48.40±4.36 | 44.39±2.37 | 38.57±5.36 | 36.22±2.63 | **57.53±0.67** |
| CITESEER | ACC | 48.54±0.54 | 57.86±0.67 | 39.25±0.85 | 63.62±1.10 | 66.00±1.47 | 64.16±1.06 | 65.74±0.56 | **70.12±0.36** |
| | NMI | 20.23±0.23 | 30.73±0.84 | 27.67±0.41 | 39.13±1.17 | 39.46±1.44 | 39.35±1.13 | 40.98±0.57 | **43.56±0.35** |
| | ARI | 13.34±0.66 | 27.13±0.45 | 12.57±0.72 | 37.09±1.73 | 38.66±2.24 | 37.78±1.43 | 39.66±0.91 | **44.85±0.69** |
| | F1 | 43.50±0.42 | 54.73±0.58 | 30.40±0.28 | 60.36±0.85 | 58.50±1.24 | 60.39±1.01 | 62.00±0.81 | **65.01±0.39** |
| UAT | ACC | 47.61±1.52 | 49.83±1.17 | 45.65±0.66 | 47.09±2.19 | 53.09±0.71 | 50.47±1.31 | 52.39±2.00 | **55.31±2.42** |
| | NMI | 21.66±1.42 | **25.46±0.62** | 18.50±1.25 | 16.79±2.90 | 22.61±0.67 | 23.20±1.43 | 23.30±1.26 | 24.40±1.69 |
| | ARI | 17.71±1.90 | 21.62±1.06 | 14.46±1.72 | 11.82±3.19 | 21.00±1.58 | 14.64±2.76 | **22.17±2.43** | 22.14±1.67 |
| | F1 | 43.01±2.30 | 47.69±1.64 | 45.58±0.80 | 45.04±2.37 | 51.32±0.86 | 50.26±2.45 | 48.81±2.62 | **52.77±2.61** |

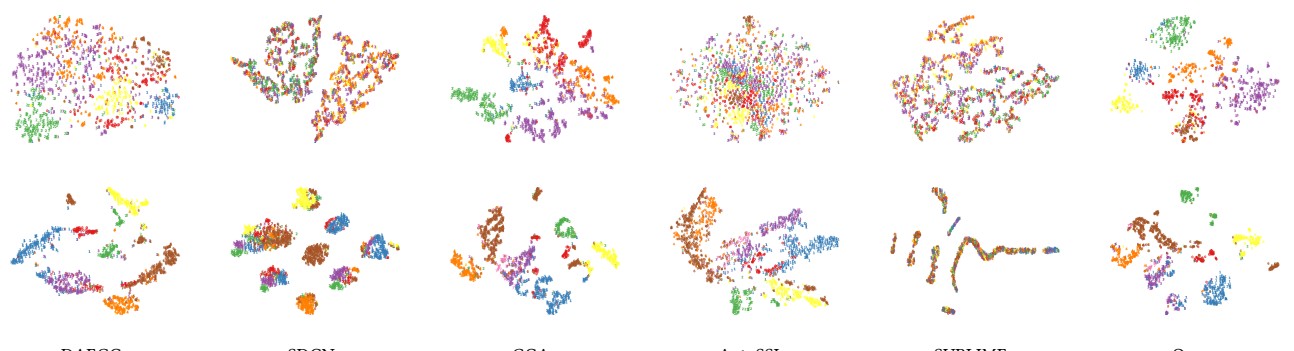

DAEGC     SDCN     GCA     AutoSSL     SUBLIME     Ours

**Figure 2: 2D $t$-SNE visualization of seven methods on two benchmark datasets. The first row and second row corresponds to CORA and AMAP dataset, respectively.**

(DAEGC [28], SDCN [1]), contrastive deep graph clustering methods (DCRN [15], AGC-DRR [7], CCGC[36], CONVERT[37]), graph structure learning methods (SUBLIME [17]), and graph augmentation methods (GCA [45], AFGRL [13], AutoSSL [11]). Moreover, due to the limitation of the space, we conduct additional comparison experiments with 5 baselines. These results are shown in Table. 2 of the Appendix. The results could also demonstrate the superiority of GraphLearner.

Here, we adopt the attention structure augmentor and the MLP attribute augmentor to generate the augmented view in a learnable way. The results are shown in Table.3 , we observe and analyze as follows.

1) GraphLearner obtains better performance compared with classical deep clustering methods. The reason is that they rarely design a specifically contrastive learning strategy to capture the supervision information implicitly.

2) Contrastive deep graph clustering methods achieve sub-optimal performance compared with ours. We conjecture that the discriminative capacity of our GraphLearner is improved with fully learnable augmentation and optimization strategies.

3) The classical graph augmentation methods achieve unsatisfied clustering performance. This is because they merely consider the learnable of the structure, while neglecting the attribute. Moreover, most of those methods can not optimize with the downstream tasks.

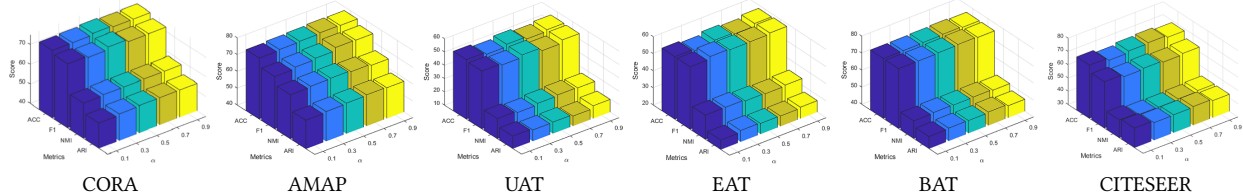

Figure 3: Sensitivity analysis of the hyper-parameter $\alpha$.

Table 6: Training Time Comparison on five datasets with six methods. The algorithms are measured by seconds. The Avg. represents the average time cost on five datasets.

| Method | CORA | AMAP | EAT | CITESEER | UAT | Avg. |
|---|---|---|---|---|---|---|
| DEC | 91.13 | 264.20 | 26.99 | 223.95 | 42.30 | 129.71 |
| DCN | 47.31 | 94.48 | 9.56 | 74.69 | 29.57 | 51.12 |
| DAEGC | 12.97 | 39.62 | 5.14 | 14.70 | 6.44 | 15.77 |
| AGE | 46.65 | 377.49 | 3.86 | 70.63 | 8.95 | 101.52 |
| MVGRL | 14.72 | 131.38 | 3.32 | 18.31 | 4.27 | 64.4 |
| SCAGC | 54.08 | 150.54 | 47.79 | 50.00 | 64.70 | 73.42 |
| **Ours** | 10.06 | 83.26 | 2.21 | 17.24 | 4.15 | 23.38 |

4) It could be observed that the graph structure learning methods are not comparable with ours. We analyze the reason is that those methods refine the structure with the unreliable strategy at the beginning of the training. In summary, our method outperforms most of other algorithms on six datasets with four metrics. Taking the result on CORA dataset for example, GraphLearner exceeds the runner-up by 1.70%, 2.38%, 2.37%, 2.00% with respect to ACC, NMI, ARI, and F1.

Moreover, we implement experiments on other augmentors. The results are shown in Table. 7, we conclude that our proposed augmentors could achieve better performance on most metrics compared with other graph clustering algorithms.

## 3.3 Time Cost and Memory Cost (RQ2)

In this subsection, we implement time and memory cost experiments to demonstrate the effectiveness of the proposed GraphLearner.

Specifically, we test the training time of GraphLearner with six baselines on six datasets. For fairness, we train all algorithms with 400 epochs. From the results in Table 6, we observe that the training time of GraphLearner is comparable with other seven algorithms. The reason we analyze is as follows: instead of using GCN as the encoder network, we adopt the graph filter to smooth the feature. This operation effectively reduces time consumption.

Moreover, we conduct experiments to test GPU memory costs of our proposed GraphLearner. Due to the limited space, we conduct the memory experiments on Fig.1 of the Appendix. The results demonstrate that the memory costs of our GraphLearner are also comparable with other algorithms.

## 3.4 Ablation Studies (RQ3)

In this section, we first conduct ablation studies to verify the effectiveness of the proposed modules. Due to the limited space, we conduct experiments about the effectiveness of the the similarity

Table 7: Clustering performance on other augmentors.

| Method | Metric | CORA | AMAP | BAT | EAT | UAT |
|---|---|---|---|---|---|---|
| **X_ATT&A_GCN** | ACC | 74.48 | 77.76 | 74.89 | 57.89 | 56.02 |
| | NMI | 54.95 | 65.97 | 50.37 | 34.07 | 26.09 |
| | ARI | 51.09 | 58.70 | 46.42 | 27.32 | 22.87 |
| | F1 | 73.79 | 69.89 | 74.97 | 58.05 | 55.49 |
| **X_ATT&A_MLP** | ACC | 73.91 | 77.58 | 72.06 | 57.22 | 55.18 |
| | NMI | 57.92 | 66.74 | 47.24 | 33.77 | 23.73 |
| | ARI | 49.58 | 58.47 | 42.48 | 27.38 | 21.92 |
| | F1 | 70.70 | 72.21 | 71.86 | 56.40 | 52.28 |
| **X_ATT&A_ATT** | ACC | 74.37 | 77.22 | 74.96 | 57.52 | 55.33 |
| | NMI | 55.30 | 66.71 | 50.04 | 33.67 | 25.17 |
| | ARI | 50.57 | 57.53 | 46.40 | 27.05 | 21.40 |
| | F1 | 73.95 | 72.65 | 75.04 | 57.62 | 55.36 |
| **X_MLP&A_GCN** | ACC | 74.70 | 77.29 | 73.44 | 56.97 | 52.04 |
| | NMI | 56.54 | 66.14 | 51.55 | 33.07 | 22.08 |
| | ARI | 50.80 | 57.04 | 47.26 | 26.05 | 17.20 |
| | F1 | 74.47 | 70.96 | 71.93 | 57.25 | 48.01 |
| **X_MLP&A_MLP** | ACC | 73.97 | 77.54 | 75.19 | 56.89 | 55.60 |
| | NMI | 55.07 | 66.59 | 50.20 | 33.48 | 24.99 |
| | ARI | 49.80 | 58.14 | 46.78 | 26.37 | 24.38 |
| | F1 | 73.55 | 72.47 | 75.25 | 57.09 | 53.97 |
| **X_MLP&A_ATT** | ACC | 74.91 | 77.24 | 75.50 | 57.22 | 55.31 |
| | NMI | 58.16 | 67.12 | 50.58 | 33.47 | 24.40 |
| | ARI | 53.82 | 58.14 | 47.45 | 26.21 | 22.14 |
| | F1 | 73.33 | 73.02 | 75.40 | 57.53 | 52.77 |

and pseudo-label matrix refinement strategies on Fig. 2 in Appendix.

*3.4.1 Effectiveness of the Structure and Attribute Augmentor.* To verify the effect of the proposed structure and attribute augmentor, we conduct extensive experiments as shown in Table 5. Here, we adopt "(w/o) $\mathbf{Aug_X}$", "(w/o) $\mathbf{Aug_S}$" and "(w/o) $\mathbf{Aug_X}$ & $\mathbf{Aug_S}$" to represent the reduced models by removing the structure augmentor, the attribute augmentor, and both, respectively. From the observations, it is apparent that the performance will decrease without any of our proposed augmentors, revealing that both augmentors make essential contributions to boosting the performance. Taking the result on the CORA dataset for example, the model performance is improved substantially by utilizing the attribute augmentor.

*3.4.2 Effectiveness of our learnable augmentation.* To avoid the existing and predefined augmentations on graphs, we design a novel fully learnable augmentation method for graph clustering. In this part, we compare our view construction method with other classical graph data augmentations, including mask feature [43], drop edges [34], add edges [34], and graph diffusion [25]. Concretely, in Table 5, we adopt the data augmentation as randomly dropping 20% edges ("Drop Edges"), or randomly adding 20% edges ("Add

Edges"), or graph diffusion ("Diffusion") with 0.20 teleportation rate, or randomly masking 20% features ("Mask Feature"). From the results, we observe that the performance of commonly used graph augmentations is not comparable with ours. In summary, extensive experiments have demonstrated the effectiveness of the proposed learnable augmentation.

### 3.5 Hyper-parameter Analysis (RQ4)

We verify the sensitivity of $\alpha$. The experimental results are shown in Fig.3. The performance will not fluctuate greatly when $\alpha \in [0.1, 0.9]$. From these results, we observe that our GraphLearner is insensitive to $\alpha$ when $\alpha \in [0.1, 0.9]$. Due to the limited space, we investigate the influence of the hyper-parameter threshold $\tau$. Experimental evidence can be found in Fig. 3 in Appendix.

### 3.6 Visualization Analysis (RQ5)

In this subsection, we visualize the distribution of the learned embeddings to show the superiority of GraphLearner on CORA and AMAP datasets via $t$-SNE algorithm [26]. We implement experiments with DAEGC [28], SDCN [1], GCA [45], AutoSSL [11], SUBLIME [17] and ours. The results are shown in Fig. 2. From the results, we can conclude that GraphLearner better reveals the intrinsic clustering structure.

## 4 RELATED WORK

### 4.1 Contrastive Deep Graph Clustering

The existing deep graph clustering methods can be roughly categorized into three classes: generative methods [1, 4, 22, 25, 28, 29, 42], adversarial methods [20, 21, 24], and contrastive methods [5, 10, 15, 16, 34, 43]. In recent years, the contrastive learning has achieved great success in vision [2, 3, 8, 41] and graph [33, 40, 44]. In this paper, we focus on the data augmentation of the contrastive deep graph clustering methods. Concretely, a pioneer AGE [5] conducts contrastive learning by a designed adaptive encoder. Besides, MVGRL [10] generates two augmented graph views. Subsequently, DCRN [15] aims to alleviate the collapsed representation by reducing correlation in both sample and feature levels. Meanwhile, the positive and negative sample selection have attracted great attention of researchers. Concretely, GDCL [43] develops a debiased sampling strategy to correct the bias for negative samples. Although promising performance has been achieved, previous methods generate different graph views by adopting uniform data augmentations like graph diffusion, edge perturbation, and feature disturbance. Moreover, these augmentations are manually selected and can not be optimized by the network, thus limiting performance. To solve this problem, we propose a novel contrastive deep graph clustering framework with fully learnable graph data augmentations.

### 4.2 Data Augmentation in Graph Contrastive Learning

Graph data augmentation [30, 31] is an important component of contrastive learning. The existing data augmentation methods in graph contrastive learning could rough be divided into three categories, i.e., augment-free methods[13], adaptive augmentation methods [39, 45], and learnable data augmentation methods [11, 23, 35, 38].

AFGRL[13] generates the alternative view by discovering nodes that have local and global information without augmentation. While the diversity of the constructed view is limited, leading to poor performance. Furthermore, to make graph augmentation adaptive to different tasks, JOAO [39] learns the sampling distribution of the predefined augmentation to automatically select data augmentation. GCA [45] proposed an adaptive augmentation with incorporating various priors for topological and semantic aspects of the graph. However, the augmentation is still not learnable in the adaptive augmentation methods. Besides, in the field of graph classification, AD-GCL [23] proposed a learnable augmentation for edge-level while neglecting the augmentations on the node level. More recently, AutoGCL[38] proposed a probability-based learnable augmentation. Although promising performance has been achieved, the previous methods still rely on the existing and predefined data augmentations. CONVERT [37] places a strong emphasis on the semantic reliability of augmented views by leveraging a reversible perturb-recover network to generate embeddings for these augmented views. However, it tends to overlook the significance of the underlying graph topology. In this work, we propose a fully learnable augmentation strategy specifically tailored for graph data. To highlight the uniqueness of our approach, we draw comparisons with existing graph data augmentation methodologies. First and foremost, GraphLearner stands out by introducing a fully learnable augmentation approach for graphs. This sets it apart from adaptive augmentation methods such as GCA [45] and JOAO [39], which still rely on predefined augmentations. GraphLearner dynamically generates augmented views using a learnable process, considering both structural and attribute aspects of the graph. Moreover, while existing learnable augmentation methods design specific adaptable strategies, they often lack task-specific adaptability.

Most of those methods are always graph classification, e.g., AD-GCL [23] and AutoGCL [38]. In contrast, GraphLearner addresses this limitation by tailoring the augmentation strategy to graph node clustering in an unsupervised scenario. This ensures that the augmentation is not only learnable but also responsive to downstream task results, enhancing its overall effectiveness.

## 5 CONCLUSION

In this work, we propose a fully learnable augmentation method for graph contrastive clustering termed GraphLearner. To be specific, we design a fully learnable augmentation with the structure augmentor and the attribute augmentor to dynamically learn the structure and attribute information, respectively. Besides, an adversarial mechanism is designed to keep cross-view consistency in the latent space while ensuring the diversity of the augmented views. Meanwhile, we propose a two-stage training strategy to obtain more reliable clustering information during the model training. Benefiting from the clustering information, we refine the learned structure with the high-confidence pseudo-label matrix. Moreover, we refine the augmented view with the cross-view sample similarity matrix to further improve the discriminative capability of the learned structure. Extensive experiments on six datasets demonstrate the effectiveness of our proposed method. In the future, it is worth trying the augmentors designed in GraphLearner for other graph downstream tasks, e.g., node classification and link prediction.

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
