# OpenReview forum: "GraphLearner: Graph Node Clustering with Fully Learnable Augmentation"
_acmmm.org/ACMMM/2024/Conference — MM2024 Poster_

### Official Review · Reviewer_oTbD · 2024-05-21

**Rating:** 5
**Confidence:** 4

**Summary:**

## **1. Summary.**
The authors aim that many graph node clustering methods always use the augmentations predefined by human experiences, which lacks the flexibility for the node clustering task. To improve the augmentation, the authors designed a learnable graph augmentation called GraphLearner.

**Strengths:**

## **2. Strengths.**
1. The motivation is well described. The authors design a learnable augmentation method to improve the node clustering performance.
2. The authors state the properties of existing augmentation methods. Different with the existing graph node clustering method, the authors propose the augmentors could learn the structure and attribute during the training.
3. The experiments are widely conducted. The experiments in this paper include the comparison experiments, hyper-parameter analysis, ablation experiments, and visualization experiments.

**Limitations:**

## **3. Some minor issues.**
After carefully reading the entire paper, it can be seen that it has been carefully prepared. I only have a few small issues to provide to the authors for further clearing.

**About experiments:** In Line 795, the authors provide many ablation experiments. The ablation experiments in Subsection 3.4.1 contain three components, i.e., (w/o) Aug_X, (w/o) Aug_S, and (w/o) Aug_S & Aug_X. However, a detailed setting should provided about this experiment for w/o Aug_X, Aug_S and (w/o) Aug_S & Aug_X. The current version lacks information. Moreover, the authors should give details about the loss function used in the first stage and the second stage. Which stage uses the loss function defined in Eq.(15)? What is the loss function used for another training stage? It makes me a bit confused. This version lacks a clear description of the loss function.

**About future work:** The author should analyze the shortcomings of the design method and the directions for future improvement and exploration in the conclusion.  This paper designs a learnable augmentation method for graph node clustering task.  Designing the structure and attribute augmentors are the advantages of this paper.  The shortcomings are not described in the paper.

**Other small details:** There are some typos in this paper, Line 868, augment-free methods can be polished as augment-free method. Typos will affect the overall readability of a paper. Therefore, the authors should carefully check the paper.

**Overall, this is a good work, motivation, method presentation, and experiments are all carefully performed. If the authors can clear some my issue to improve the quality of this paper, I am willing to increase my rating.**

**Suitability:**

3

---

### Official Review · Reviewer_EQMz · 2024-05-23

**Rating:** 5
**Confidence:** 4

**Summary:**

This paper proposes a graph node clustering method, named GraphLearner, which enhances contrastive graph node clustering by introducing a learnable structure and attribute augmentor, as well as a dual refinement module. The dual refinement module refines the augmented graph using a cross-view similarity matrix and a pseudo-label matrix.

**Strengths:**

1. GraphLearner introduces a learnable way to augment the structure of the augmented graph, significantly improving contrastive graph node clustering performance compared to baselines that use predefined augmentations or only augment attributes in a learnable manner.
2. GraphLearner refines the augmented graph, making it more suitable for downstream clustering tasks.
3. The manuscript is well-organized, making it clear and easy to understand.
4. The experiments are generally comprehensive and the results are good.

**Limitations:**

1. The introduction highlights that previous learnable augmentation methods neglect the learning of effective attribute augmentations. However, an experimental comparison of CCGC and CONVERT, which only augment attributes in a learnable way, with the proposed GraphLearner (when the structure augmentation is ablated) in Table 5, reveals that the attribute augmentation performance of GraphLearner might not be satisfactory.
2. There is a lack of explanation regarding how the experiments ablating both structure and attribute augmentations were conducted. If neither attribute nor structure is augmented, how is the contrastive learning carried out?
3. The Related Works section of the manuscript needs improvement. For instance, the involved baseline AutoSSL is not introduced in the manuscript.

**Suitability:**

2

---

### Official Review · Reviewer_id1q · 2024-05-26

**Rating:** 4
**Confidence:** 3

**Summary:**

CDGC methods utilize contrast learning to group nodes, and high-quality contrast samples are critical for performance. Existing methods' enhancement samples are predefined based on human experience and irrelevant to downstream clustering tasks, leading to high labor costs and poor performance. GraphLearner is proposed to introduce a learnable enhancer to generate high-quality, task-relevant enhancement samples. GraphLearner contains two learnable enhancers for capturing attribute and structural information. A high-confidence pseudo-labeling matrix and a cross-view sample similarity matrix are introduced to enhance the reliability of the learned affinity matrix. During the training process, the difference in optimization objectives between the learning enhancer and the comparison learning network is noted, which needs to ensure the consistency of the embedding and the diversity of the enhancement samples, for which the adversarial learning mechanism is proposed.

**Strengths:**

1. It proposes the GraphLearner framework, a fully learning data enhancement method for deep comparative graph clustering that dynamically learns structural and attribute information.
2. Optimizes the augmented graph structure using a cross-view similarity matrix and a high-confidence pseudo-labeling matrix to improve the reliability of the learned affinity matrix.
3. integrate the clustering task and augmented learning into a unified framework that mutually reinforces each other.

**Limitations:**

1. The experimental dataset is insufficient and should be tested on larger graphs, e.g. ogbn-papers 100M, ogbn-products.
2. The Structure Augmentor and Attribute Augmentor mentioned in the Fully Learnable Augmentation Module module mentioned in the paper are not very generalizable and must be adapted for different network frameworks. I can understand that different backbones require special handling, but the runtime logic between MLP-based, GCN-based, and Attention-based may not be uniform enough. Also, I doubt these three methods will cover all cases, such as APPNP.
3. The lack of robustness of the model may also be a problem, and in Figure 3, we see that the sensitivity of the hyper-parameter α is too strong.

**Suitability:**

2

---

### Official Review · Reviewer_wdih · 2024-05-27

**Rating:** 4
**Confidence:** 3

**Summary:**

This paper introduces a novel approach to contrastive deep graph clustering (CDGC) by employing fully learnable augmentations for generating high-quality, task-specific samples. It incorporates two learnable augmentors for attribute and structural information, and introduces high-confidence pseudo-label and cross-view sample similarity matrices to refine the learned affinity matrix. An adversarial learning mechanism ensures consistency and diversity, while a two-stage training strategy further enhances clustering reliability. Extensive experiments on benchmark datasets show that GraphLearner outperforms existing methods, demonstrating superior clustering accuracy and robustness.

**Strengths:**

1. The paper introduces fully learnable augmentations for contrastive deep graph clustering, which is a significant advancement over traditional predefined augmentations. This dynamic generation of high-quality, task-specific samples enhances clustering performance.

2. The integration of high-confidence pseudo-label and cross-view sample similarity matrices to refine the learned affinity matrix is a novel approach that improves the reliability and discriminative power of the clustering process.

3. Extensive experiments on multiple benchmark datasets demonstrate that GraphLearner significantly outperforms existing state-of-the-art methods. The results validate the effectiveness and efficiency of the proposed approach in various graph node clustering scenarios.

4. The paper is well-structured and clearly explains the proposed methods, including detailed descriptions of the learnable augmentors, adversarial learning mechanism, and two-stage training strategy. This clarity aids in the reproducibility and understanding of the research contributions.

**Limitations:**

1. The proposed fully learnable augmentation strategy and adversarial learning mechanism add significant complexity to the implementation. This might make it challenging for practitioners to adopt and integrate the method into existing systems without substantial effort.

2. While the paper demonstrates effectiveness on benchmark datasets, it does not extensively discuss the scalability of the approach for very large graphs. The computational and memory requirements for the proposed method could be a limitation for large-scale applications.

3. Typos: Figure 1 caption: "we design the learnable augmentors to to dynamically learn the structure and attribute information." (Repeated "to")

**Suitability:**

2

---

### Meta-Review · Area_Chair_HYqn · 2024-06-27

**Recommendation:** Accept (Poster)
**Confidence:** 5

**Metareview:**

According to all the review comments, rebuttals, discussions and final ratings, the majority of the reviewers gave positive ratings to this paper and the concerns were well addressed. I am happy to recommend to accept this paper. Please carefully revise the final manuscript according to the comments and discussions.